# A locally-blazed ant trail achieves efficient collective navigation despite limited information

Ehud Fonio[1], Yael Heyman[1], Lucas Boczkowski[2], Aviram Gelblum[1], Adrian Kosowski[3], Amos Korman[2]*, Ofer Feinerman[1]*

[1]The Department of Physics of Complex Systems, Weizmann Institute of Science, Rehovot, Israel; [2]Institut de Recherche en Informatique Fondamentale, CNRS and University Paris Diderot, Paris, France; [3]Institut de Recherche en Informatique Fondamentale, INRIA and University Paris Diderot, Paris, France

**Abstract** Any organism faces sensory and cognitive limitations which may result in maladaptive decisions. Such limitations are prominent in the context of groups where the relevant information at the individual level may not coincide with collective requirements. Here, we study the navigational decisions exhibited by *Paratrechina longicornis* ants as they cooperatively transport a large food item. These decisions hinge on the perception of individuals which often restricts them from providing the group with reliable directional information. We find that, to achieve efficient navigation despite partial and even misleading information, these ants employ a locally-blazed trail. This trail significantly deviates from the classical notion of an ant trail: First, instead of systematically marking the full path, ants mark short segments originating at the load. Second, the carrying team constantly loses the guiding trail. We experimentally and theoretically show that the locally-blazed trail optimally and robustly exploits useful knowledge while avoiding the pitfalls of misleading information.

*For correspondence: ofer.feinerman@weizmann.ac.il (OF); amos.korman@irif.fr (AKor)

**Competing interests:** The authors declare that no competing interests exist.

## Introduction

Decision-making using noisy sensory information is a well-studied topic (*Kochenderfer, 2015*; *Chittka et al., 2009*; *Kepecs et al., 2008*). However, even in the absence of noise, cognitive or sensory limitations filter those aspects of the environment that are perceived by an organism (*Von Uexkull, 1957*). Relying on partial knowledge may result in maladaptive decisions (*Schlaepfer et al., 2002*), where even a single bad decision somewhere among a long sequence of good ones has the potential to lead along an undesirable path (*von Herrath et al., 1994*; *Davies, 2000*). This difficulty is pronounced in the cooperative contexts of multi-cellular systems or animal groups (*Czaczkes et al., 2016*; *Schmidt et al., 2006*; *Beckers et al., 1990*; *Braess, 1968*) which collectively require information that may be inaccessible to their individual members (*Berdahl et al., 2013*; *Robinson et al., 2014*).

Cooperative transport by ants is a behavior in which ants join to collectively carry an object that is too heavy for any single one of them (*McCreery and Breed, 2014*; *Czaczkes and Ratnieks, 2013*; *Gelblum et al., 2015*). The inherent scale differences that characterize this behavior hold the potential of introducing conflicts between the knowledge available to individual ants to that required for the load's navigation. This system provides a unique opportunity for studying the consequences and possible resolution of such conflicts due to two main reasons. The first is experimental tangibility: In the context of cooperative transport, abstract decision making processes are manifested as collective motions; Sequences of decisions are evident as the load's direction changes wherein bad

**eLife digest** Ants forage to find food and bring it back to the colony. If they come across food items that are too large or heavy for a single individual to carry, some species are able to form teams to cooperatively carry these items to the nest. This collective navigation process hinges on the navigational abilities of the individual ants. However, in natural terrains, the routes that are available to an individual ant are often inaccessible for a large group carrying a bulky item. So how do the ants manage to navigate together?

Fonio et al. studied how longhorn crazy ants cooperate to move large items. The experiments show that nearby ants not currently engaged in carrying the item mark the ground with chemical scents. Fonio et al. devised an automated method of detecting scent marking events and this has provided some of the first real time documentation of ant scent trails as they form. This shows that when cooperating to move large objects, the ants use scent marks to form a new type of trail that is highly dynamic. Unlike other ant trails that mark the whole path between the food and the nest, these new trails only direct the next step of the movement. Furthermore, the team of ants carrying the item only follows these local directions in a loose manner and often ignores them.

Fonio et al. then used a mathematical model and further experiments to show that this new type of trail effectively solves the problems of collective navigation during cooperative transport. Essentially, the locality of the trail and the loose way in which the group follows it tune the degree to which the collective motion depends on the directions provided by individual ants. This allows the group to benefit from the useful information available to individuals while avoiding local traps that may occur when these individuals wrongly direct them towards dead ends.

The next step following on from this work is to understand the mechanisms behind this newly discovered trail, and in particular, understand how the collective motion results from the actions of individual ants that react to single drops of scent. Another challenge for future research would be to find technological applications for this newly discovered strategy, such as routing over communication networks.

turns eventually lead toward dead-ends. Second, cooperative transport is highly efficient as a general transportation solution. Ants manage to maneuver a wide range of loads while quickly navigating through complex terrains and back to their nest (*Czaczkes and Ratnieks, 2013*; *Gelblum et al., 2015*).

Here, we study the cooperative transport by *P. longicornis* ants (*McCreery and Breed, 2014*; *Czaczkes and Ratnieks, 2013*; *Gelblum et al., 2015*), an invasive species with a broad worldwide distribution (*Wetterer, 2008*). Similar to other ant species, *P. longircornis* employ a pheromonal recruitment mechanism (*Czaczkes et al., 2013b*). Here, we demonstrate, for the first time in any ant species, the use of scent marks that assist navigation during the retrieval stage of cooperative transport. Tracking these scent marks provides us with a direct glance into the relations between the information that individual ants present to the group and the subsequent navigational choices on the collective scale.

We find that while the information conveyed by scent marks is typically precise, it is occasionally misleading. This is because individual ants appear to be ignorant of the accessible paths from the perspective of the much larger load. When grouped together, the scent marks collectively left by the ants form a *locally-blazed* trail that is characterized by short range markings and highly stochastic following. We discuss the traits of this trail in the context of previously described trails and suggest that it should be considered as a new kind of ant trail. We further use an algorithmic approach to show how the unique characteristics of the locally-blazed trail are suited to the distinct navigational challenges of cooperative transport. To support our theoretical and numerical findings, we present experimental evidence for efficiency in the ants' collective motion along with experimental verification of some of the model's predictions.

## Results

### Identification of pheromone deposition events

In *P. longicornis*, an ant that finds a food item that is too heavy to carry alone, returns to the nest and recruits help (*Figure 1A*, *Figure 1—figure supplement 1*) (*Stanley and Robinson, 2007*; *Trager, 1984*; *Kenne et al., 2005*). This recruitment trip spans the full distance to the nest (N = 12 out of 12 ants at six meters, 2000 folds the body length of the ants) and is characterized by a highly punctuated motion (*Figure 1B*) that was previously associated with pheromone laying behavior (*Holldobler and Wilson, 1990*; *Beckers et al., 1992a*; *Hölldobler et al., 1978*). Using a side-view camera we could indeed relate dips in the ant's speed with marking events in which the ant touches the tip of its gaster to the surface, a known property of marking behavior in various ant species (*Beckers et al., 1992a*; *Hölldobler et al., 1978*; *Czaczkes et al., 2013a*; *Holldobler, 1981*) (*Figure 1C*, *Video 1*). Chemical analysis of the marked surfaces ascertains that the identified events are accompanied by the deposition of undecane, a prominent *P. longicornis* trail pheromone (*Morgan et al., 2005*; *Witte et al., 2007a*; *Witte et al., 2007b*) (*Figure 1D–E*, Materials and methods). The fast evaporation of this short hydrocarbon dictates an extremely short lifetime (*Czaczkes and Ratnieks, 2013*; *Fujiwara-Tsujii et al., 2006*) in comparison to most other pheromones found in the trails of other ant species (*Morgan, 2009*; *Jackson et al., 2006*; *Beugnon and Déjean, 1992*) and even to trails constructed by *P. longicornis* outside the context of cooperative transport (*Witte et al., 2007a*). In fact, the high volatility of undecane makes it a prevalent ant alarm pheromone (*Fujiwara-Tsujii et al., 2006*; *Blum, 1969*; *Lenz et al., 2013*; *Regnier and Wilson, 1968*). The behavioral response of *P. longicornis* to this pheromone has, accordingly, been demonstrated to elicit short lived attractive responses, on the order of one minute (*Witte et al., 2007b*).

The association between a specific speed signature and the lowering of the gaster enables the identification of marking events using a top view camera (93% true positives and a false-detection-rate of 13.1%, see *Figure 1—figure supplement 2*, *Video 2* and *Video 3*). This holds a significant advantage since discerning marking events by use of a top-view, rather than a side-view, increases the area over which this behavior can be tracked from a narrow one-dimensional bridge (*Beckers et al., 1992*; *Aron et al., 1989*; *Beckers et al., 1993*) to a large two-dimensional surface. Using this technique, we pinpoint the details of the ants' dynamical scent map over the large length scales relevant for navigation by *P. longicornis* ants (*Figure 1F*, *Video 4* and *Video 5*).

### Scent marks in error-prone environments

A prominent characteristic of cooperative transport in *P. longicornis* ants is that a large fraction of ants accompany the load rather than physically carry it (*Czaczkes et al., 2013b*). Our detection method reveals that many of these ants engage in marking behavior (*Figure 1F*), collectively laying scent marks with a mean rate of 1.4 marks per second (*T = 785* s) in the vicinity of the load. We found that when ants were prevented from obtaining information about the location of the nest (see the Experimental setup subsection of the Materials and methods), marking rates were significantly ($p < 10^{-4}$ unpaired T-test, *T = 978* s) lower at 0.001 marks per second. This suggests that scent marks convey navigational information. Indeed, we find that, in obstacle free environments, markings hold accurate information ($1.39 \pm 0.02$ bits per mark, *N = 1395*, see Materials and methods) regarding the direction to the nest (*Figure 2A*).

As may be expected, scent marks influence the load's motion: We find that, on average, each new mark transfers $0.35 \pm 0.02$ bits of directional information to the team of carriers, information that causes the group to divert the load's motion towards the mark (*Figure 2B–C*, see Materials and methods). To obtain a rough quantitative estimate on the rate of information transfer we can attribute ten such scent marks (that occur, on average, within about $10 marks/1.4$ (marks/second) = 7.1 s) with $10 \cdot 0.35 = 3.5$ bits. Since each bit reduces uncertainty by a factor of two; 3.5 bits could be enough to increase the precision of the load's direction of motion from complete lack of orientation (360°) to an accuracy of $360°/2^{3.5} = 32°$. Similarly, twenty marks occurring on a much shorter time scale than the duration of retrieval (which typically lasts many minutes) convey enough information for a high directional accuracy of under 5°.

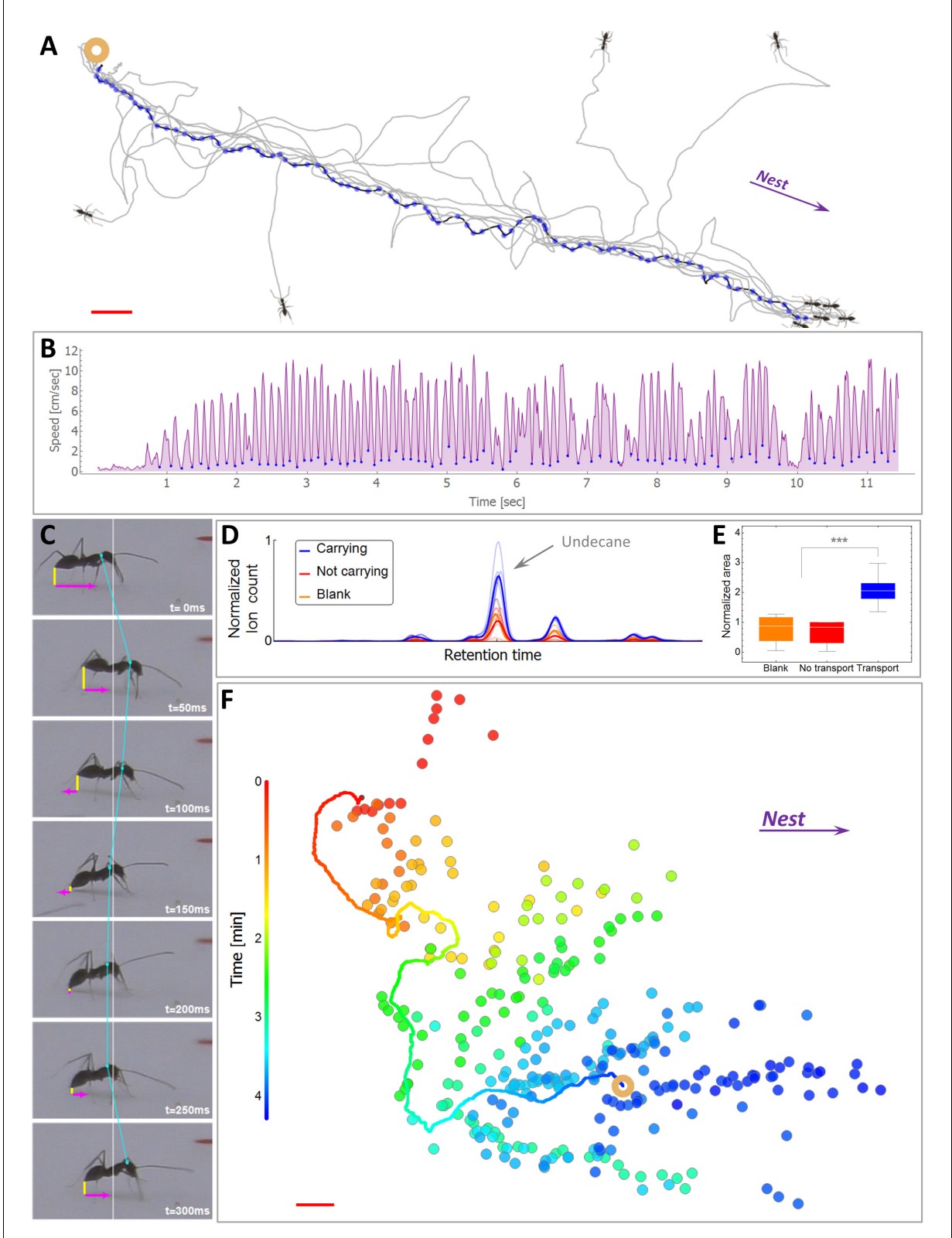

**Figure 1.** Scent mark detection. (**A**) A typical recruitment run. The first recruiter deposits a sequence of pheromone marks (blue dots) that span an entire path between the food (ring) and the nest. Briefly thereafter, other ants are recruited (see *Figure 1—figure supplement 1*) either from the surrounding area or from the nest, showing high attraction to the original scent trail. The trajctories of several such recruited ants, as they move towards the food item, are depicted by black lines. (**B**) The speed time-series of the recruiting ant depicted in panel (**A**). Blue dots indicate marking events. (**C**) A

*Figure 1 continued on next page*

*Figure 1 continued*

side view of a typical marking event characterized by both a lowering of the gaster (shortening of the vertical yellow line) and reversal in the ant's speed (magenta). See also *Figure 1—figure supplement 2* and *Video 2*. (D) Normalized ion count Gas-Chromatography-mass-Spectrometry in the region of undecane for marking ants, control ants and blank. (E) A box-plot representation of the area beneath the undecane peak displayed in (D). Asterisks denote p-value<0.0001 Kolmogorov-Smirnov test for non-equal areas between transport samples (N = 6 experiments) and grouped blank and control samples (N = 12 experiments). (F) Marking positions during cooperative transport. The marks were produced by multiple ants over a 1200 cm² area during 265 s. Full line denotes the trajectory of the load (ring) during the same time period. Color codes for elapsed time since the beginning of motion. Red bars denote 2 cm.

The following figure supplements are available for figure 1:

**Figure supplement 1.** Recruitment.

**Figure supplement 2.** Verification of marking identification.

While the high directional accuracy of scent marks is typically advantageous, it may occasionally play a negative role in ragged environments which naturally contain obstacles. Near an obstacle, markings that are directed towards the nest may form misleading trails that are inaccessible to large loads (*Figure 2D*). This could arise from the inherent gap between the perspective of an ant (*Von Uexkull, 1957*) and the relevant information that is required by the group (*Video 6*). In this case, high fidelity to the directionally accurate yet navigationally misleading information could summon trap-like conditions and deadlocks (*Schmidt et al., 2006*; *Beckers et al., 1990*) similar to those observed in classical, mass recruitment ant trails (*Czaczkes et al., 2016*; *Planqué et al., 2010*; *Beckers et al., 1990*).

## The locally-blazed ant trail

Surprisingly, despite the fact that the ant-load system generally follows the scent trail (*Figure 2B–C*), we observe that it readily abandons it (*Figure 3A*) on a length scale that is as short as 10 cm (*Figure 3B*). This behavior significantly deviates from classical trail following behavior as exhibited by numerous ant species as well as *P. longicornis* itself in other scenarios (see *Figure 3B*, *Figure 3—figure supplement 1*). Indeed, ants wander off both the classical trail (*Beekman and Dussutour, 2007*; *Deneubourg et al., 1983*) and the locally-blazed trail. However, *Figure 3B* demonstrates the stark difference between these two cases: Of the *P. longicornis* ants walking along a 'classical trail', at most 10% of them lose the trail which remains stable over time (up to several days) and over long distances. Conversely, in the locally-blazed trail the probability that the moving load approaches a scent mark continuously diminishes with distance so that segments of the trail marked even slightly away of the object become practically irrelevant to subsequent movement. We note that this observation is in agreement with the fast evaporation of undecane (*Witte et al., 2007b*). At first glance, abandoning the trail, especially the initial well-trodden recruitment trail which marks the direct path to the nest, after such short distances, seems to be counter intuitive and even mal-adaptive.

A possible hypothesis that could resolve this puzzling behavior is that the partial fidelity of the carrying team to scent marks is useful for avoiding obstacle-imposed deadlocks, as described above. To test this hypothesis, we simulated the presence of natural obstacles (*Figure 2D*, *Video 6*) by constructing a barrier which has a slit at its center that easily allows for passage of ants but is too narrow for the load. This configuration results in markings that guide the load towards the dead-end (*Figure 3C*, green marks). We find that rather than being caught in a deadlock, the transported load leaves the misleading scent marks and

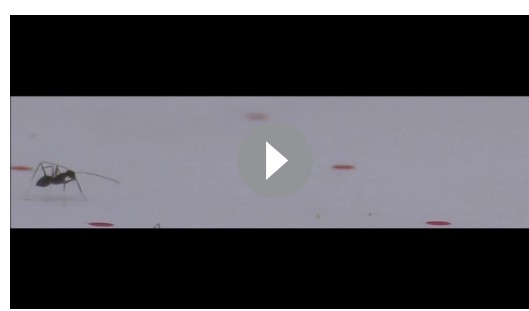

**Video 1.** Marking behavior. A side close-up view of a recruiter ant as she lays three pheromone marks. Movie slowed down four times.

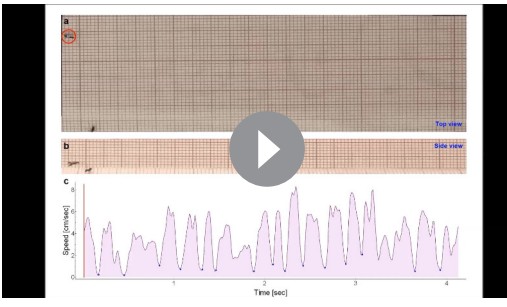

**Video 2.** Verification of marking identification. Ants were recorded from both top (a) and side (b) view simultaneously. The detection of the marking behavior, based on the ants speed profile (c) that was extracted from the top-view video, was then verified by observing frames at which the tip of the ant's gaster touches the surface as visible in the side view video.

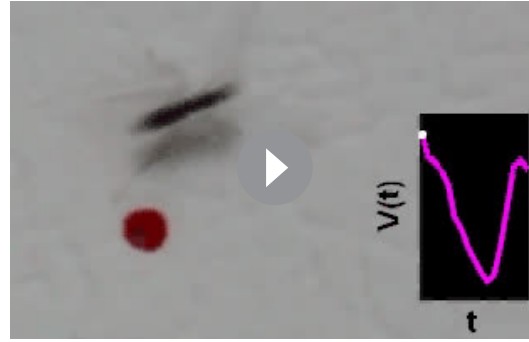

**Video 3.** Verification of marking identification including speed signature. Speed profiles that were determined as candidates for marking events were further confirmed by a human observer against the raw video, where more subtle movements that are lost in the automatic tracking can be detected. This video shows a single marking event slowed down three times.

commences a perpendicular motion along, and eventually around, the obstacle (*Figure 3C*). This effect of stochasticity is, in some sense, reminiscent of the positive role held by individual ants that wander off classical ant trails (*Beekman and Dussutour, 2007*; *Deneubourg et al., 1983*) (more about this point in the Discussion below).

The fact that the load is prone to leave the scent trail reduces the probability that it will approach scent marks laid far from it (*Figure 3B*). In light of this unpredictable motion, investing in the construction of a long trail that connects the moving load to the nest appears to be wasteful. Indeed, we find that the vast majority of individual ants deposit markings in bouts which originate near the load (*Figure 3—figure supplement 2*), are directed towards the nest (*Figure 2A*) and terminate a short distance away (*Figure 3D*) much before they get anywhere near the nest (which is several meters away). This locality of scent laying, with a median bout length of less than 10 cm (95% confidence interval is [8.6 cm, 10.3 cm] as estimated by a non-parametric Binomial-based method over N = 735 marking bouts), stands in stark contrast to the 600 cm trail laid by the first recruiter ant (see the section titled *Identification of pheromone deposition events*, above) ($p < 10^{-6}$ probability that first ant marking distances come from the distribution depicted in *Figure 3D*). Whereas the function of the latter is long range recruitment (*Figure 1—figure supplement 1*) the former is required both for orientation (*Figure 2A–C*) and, possibly, local recruitment of nearby ants (*Czaczkes et al., 2013b*; *Hölldobler et al., 1978*; *Holldobler, 1971*; *Schatz et al., 1997*).

Macroscopically, this individual marking and collective following behaviors lead to the formation of the locally-blazed trail; this trail significantly differs, in several aspects, from other previously described ant trails. First, rather than designating the whole route between the food and the nest (*Hölldobler et al., 1978*; *Cammaerts and Cammaerts, 1980*) (or any other two stable points that are, in no sense, required to be proximal [*Latty et al., 2011*]); the locally-blazed trail is segmented and continuously formed in the vicinity of the object such that it implies only the next step to be taken. This locality holds on the collective level that takes into account all marking ants and is very

**Video 4.** Dynamic scent map of first recruiting ant. This video depicts the first ant that found the food and initiated recruitment back to the nest. Purple discs pinpoint the times and locations of the scent marks deposited by this ant. The radius of the discs is proportional to the time that had passed since the marking event. This video is slowed down four times.

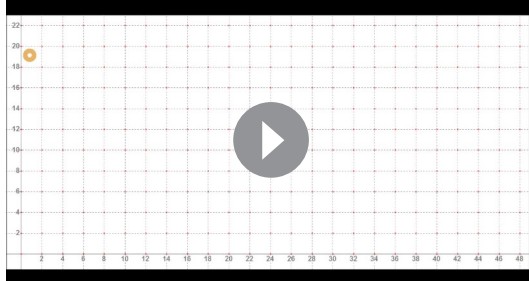

**Video 5.** Dynamic scent map during recruitment and cooperative transport. An animation, based on actual scent-marks data, of a recruitment phase followed by cooperative transport. For clarity, this video depicts the load and laid marks (as in *Video 4*) but not the participating ants. As in all other recruitment occurrences, the recruiting ant marked all the way towards the nest in a relatively straight line. In this video a mobile camera was used, and the frame was restricted by the field of view of the camera, showing an area of 20 cm × 15 cm around the load, at any given moment. All marks by all ants within this field of view are depicted in the animation, with the radius of the disc specifying the elapsed since the marking event. Conversely, any marking behavior that occurred outside the frame is not represented. The video is sped up twelve times.

different from the classical long-range punctuated or intermittent trail marking by individual ants as previously described (*Beckers et al., 1992*; *Aron et al., 1989*; *Cammaerts and Cammaerts, 1980*; *Hangartner, 1969*). Second, as the load occasionally deviates from the trail another trail dynamically forms in front of it. This happens in open space (*Figure 3A*) as well as near an obstacle (*Figure 3C*). Interestingly, the length scales of the deviation from the scent trail and the marking bout length both match and are on the order of 10 cm, possibly reflecting an evolutionary dependency between the two.

## Fault-tolerant routing

It is well known that greedy algorithms, *i.e.* algorithms that tend to optimize the next local step rather than its future consequences, often fail by converging onto a local rather than global solution. In some cases, such problems can be resolved by actively adding a random component to the algorithmic rules (*Fonseca and Fleming, 1993*; *Vermorel and Mohri, 2005*; *Kirkpatrick et al., 1983*). In this section, we explore the possibility that the locally-blazed ant trail is one of these cases.

To gain insight on the efficiency of the locally-blazed ant trail, we developed an abstract routing model (see *Supplementary file 1-1,2*). To model the terrain we discretize it into a graph. We use the freedom in determining the edge lengths of the graph such that they correspond to a reasonable step size. We naturally choose this scale to coincide with the motion's persistence length, *i.e.* the length scale over which the motion can be approximated by a straight line, in the absence of scent marks, which was previously measured to be on the order of 10 cm (*Gelblum et al., 2015*). It is the goal of a memoryless agent that has no sense of orientation to advance in a given direction as fast as possible. Inspired by the ants' behavior, our model includes two intertwined components: Marking the graph with pointers (advice) that are biased towards the desired direction and utilizing this advice to progress the navigating agent.

To model the fact that the trail is composed of broken segments (see *Figure 3D*) we anchor an advice pointer to each node. The length of a pointer is taken to be the experimentally measured median marking distance. Since this distance agrees with the grid spacing (10 cm), advice pointers can be understood to connect each node to one of its nearest neighbors. To model the unreliability of scent marks resulting from complex and unpredictable topologies, we assume that, although advice pointers typically point to the correct neighboring node, they sometimes point at an arbitrary direction.

The second part of the model describes the partial responsiveness of the load to scent marks (*Figure 3A–C*). This is modeled by the *Probabilistic-Following* algorithm: When reaching a node, the agent follows the local advice with some fixed probability and performs a random walk otherwise. These following rules have the potential of using some of the information held by the advice, while avoiding possible deadlocks in areas with misleading directions.

Adapting results from the theory of *Random Walks in Random Environments* (*Drewitz et al., 2014*; *Sznitman, 2002*), we prove that our model allows for asymptotically optimal (*i.e.* linear speed) travel times in 2D graphs (which are a good description of the world our ants live in) provided that advice error rates are sufficiently small (see *Supplementary file 1-4*). In other words, on these graphs and in the presence of occasionally misleading advice, adding noise to the advice following rules is not only *necessary* but also *sufficient* for efficient navigation. We further prove that similar results hold on line-graphs (*Supplementary file 1-3* and *Figure 4A* and *Figure 4—figure*

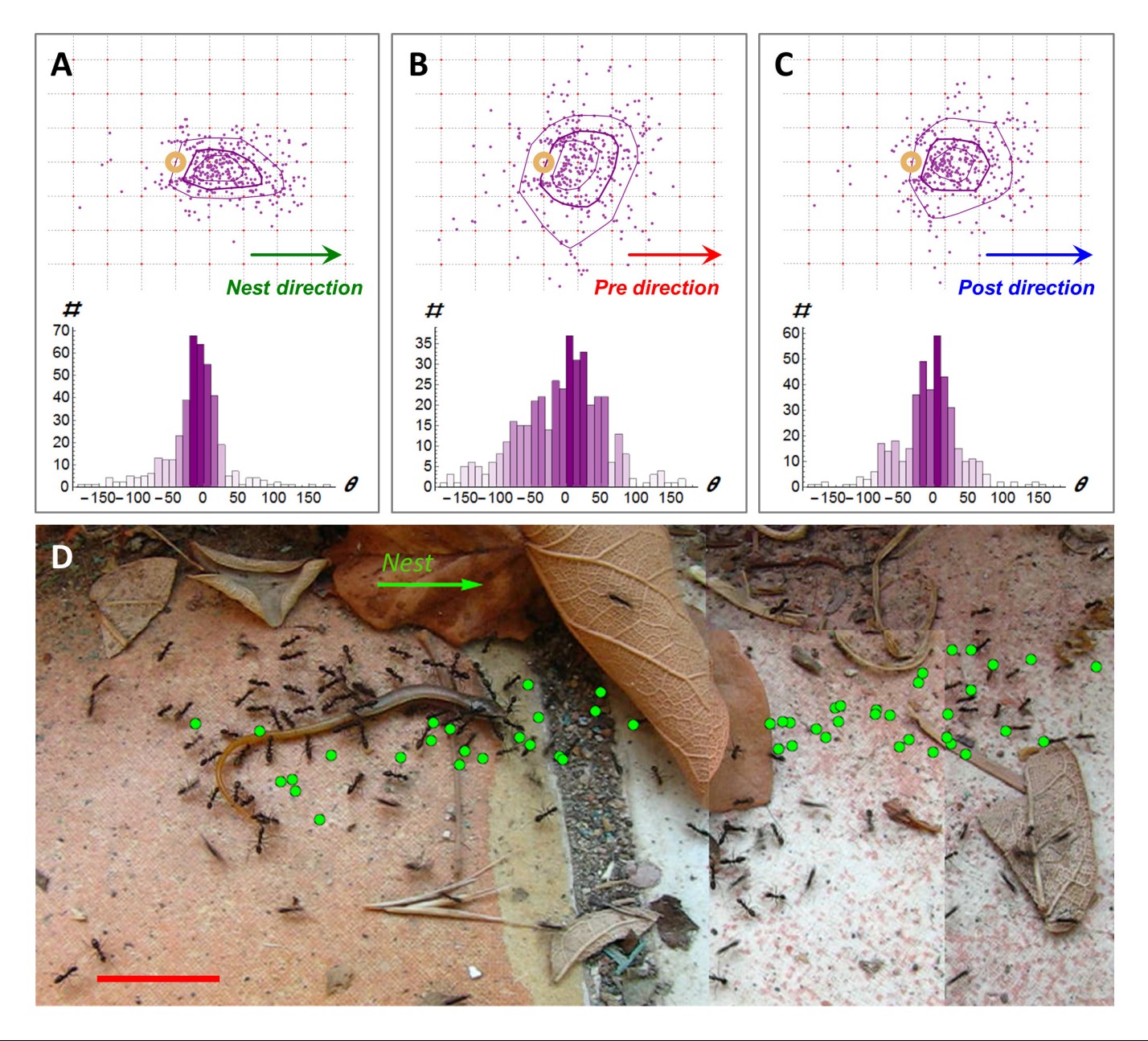

**Figure 2.** Following scent marks. (A-C). Distribution of marking events (*N = 408*) during a specific example of 113 cm of cooperative transport. Upper panels show the spatial distribution of scent marks (purple dots) locations, upon marking, in a moving frame of reference that is attached to the center of the transported load. The x-axis of this reference frame points towards the nest (a) or in the direction of the load movement in the 2 s that immediately proceed (b) or follow (c) the time this mark was deposited. Purple lines indicate quartile polytopes. Bottom panels: Angular distribution of the same data points. (D) Cooperative transport of a large prey item in a natural environment. Green dots denote scent marks. Red bars denote 2 cm.

supplement 1) and grids of all dimensions (*Supplementary file 1-4*). The question whether such

results can be obtained for the general graph case seems to be beyond the scope of currently known techniques (see Remark 23 in *Supplementary file 1-4*). It remains open for further theoretical study.

The navigational scheme described above is highly efficient at extracting useful information present in the environment even when it is masked by occasional misleading advice. Accordingly, *Probabilistic-Following* is expected to fail when most of the information is misleading. In fact, using this

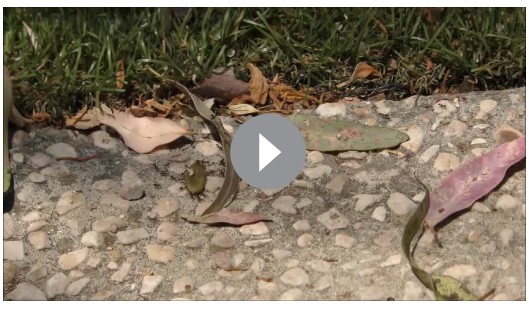

**Video 6.** Scale differences between ants and load. Ants that cooperatively carry a large seed encounter a fallen leaf. The ants can easily pass over and under the obstacle while the carried load cannot.

navigation scheme to cross large areas that contain *only* misleading information would take an exponentially long time (see *Supplementary file 1-3* and *Figure 4A*). In this case, completely ignoring advice and performing a simple random walk instead is a better strategy that yields quadratic crossing times.

## Robust routing

In an environment in which all advice is precise it is clear that the best navigational policy is to strictly follow this advice. However, in an environment which contains areas of misleading advice, this policy quickly breaks down and can lead to infinite deadlocks. In the previous section, we have shown how *Probabilistic-Following* can restore linearity in this case. It is therefore clear that an optimal navigational policy must take into account the statistics of the relevant environment. In this section, we discuss the degree to which *Probabilistic-Following* must be tuned to environmental statistics.

We prove, in *Supplementary file 1-4*, that to achieve linear passage times on the line graph there is no requirement for any sort of fine tuning. In fact, the fraction of times at which *Probabilistic-Following* ignores advice can be chosen from a large range of parameters without lengthening the travel time beyond linear. This suggests a degree of robustness in the performance of *Probabilistic-Following*.

While linearity of travel time in distance traveled is crucial it is important to verify that the coefficient of this linear relation is not too high. We define the stretch of a path on a graph as its length normalized by the length of the shortest possible path between its two end-points. We used simulations to calculate the stretch of path generated by *Probabilistic-Following* on the 2D grid. These simulations include two parameters: the fraction of correct advice (advice reliability) and the following probability of the navigational algorithm. We find that the stretch is very low (under 3) over a large fraction of the parameter space which spans following probabilities between 0.5 and 0.9 (*Figure 4B*). This implies that for a given environment, there is a large range of following probabilities that perform almost equally as well as the best possible following probability (*Figure 4C*). Furthermore, fixing the following probability to a single value within this range yields near optimal performance for almost every value of the advice reliability (see *Figure 4D*). To summarize, *Probabilistic-Following* is highly robust in the sense that it achieves near-optimal performance with hardly any requirement to fine-tune its parameters to the statistics of the environment. Such robustness and the generality which it implies are biologically appealing.

## Optimality of the locally-blazed trail

The ants' navigation algorithm shares a number of traits with our model. It is therefore interesting to check whether the navigational optimality implied by the model indeed holds in the case of ant cooperative transport.

The main prediction of the model is that travel times are linear and therefore near-optimal even in the presence of obstacles and misleading information. In the absence of obstacles, we have previously shown that, indeed, the load travels almost ballistically (*Gelblum et al., 2015*). To check if this efficiency carries over to more complex environments, we studied trail formation and load motion around an obstacle with a slit located at its center (*Figures 3C* and *5A*). In such cases, scent marks near the slit are misleading while those closer to the obstacle's edges are beneficial (*Figure 5B*). We further observe that the motion of the load around the obstacle exhibits partial responsiveness to the underlying scent marks (see the trajectory as depicted in *Figure 5B*) thus validating the assumptions of our model.

The partial responsiveness of the ant-group to the scent marks entails two different effects: On the one hand as discussed above, despite misleading marks near the slit, partial responsiveness

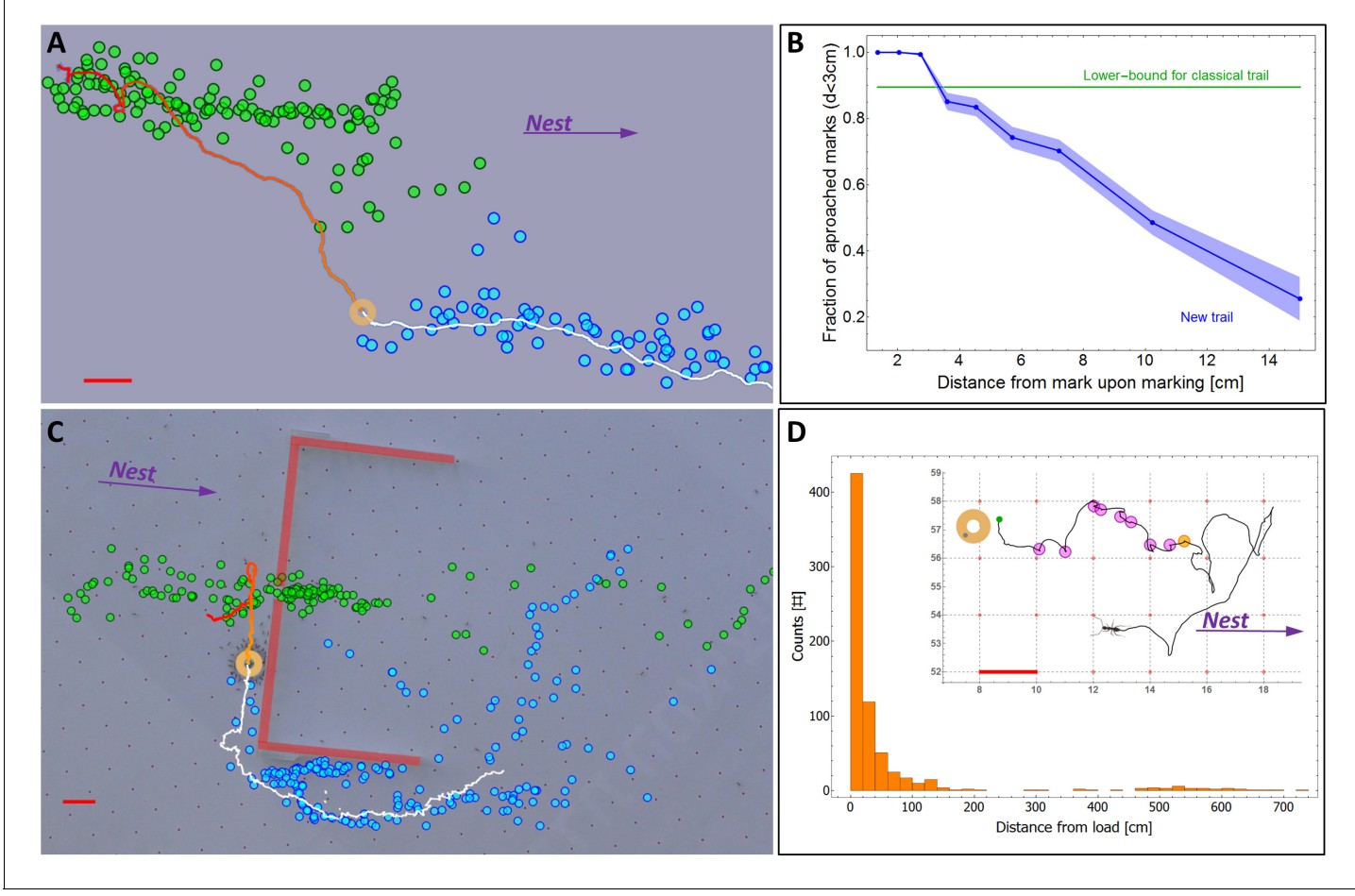

**Figure 3.** A new kind of ant trail. (A) In an obstacle-free environment, the load loses the scent trail which then reforms in front of it. Green dots indicate the position of scent marks produced before the load reached the position indicated on the image (ring). Blue dots indicate scent marks laid after this time. Solid line marks the load's trajectory. (B) The probability that the load eventually approaches (to less than 3 cm) a scent mark as a function of the distance between them at the moment of marking. For comparison, the green line depicts the corresponding curve for a classical ant trail (see *Figure 3—figure supplement 1*) (C) Cooperative transport while bypassing an obstacle (thick red lines) with a slit. Load position and marking colors as in panel (a). (D) Distribution of single ant marking bout lengths defined as the distance between the load and the furthest mark in a marking sequence. The inset shows a typical marking bout of nine marks (discs). Furthest mark is denoted in orange. Red bars denote 2 cm.

The following figure supplements are available for figure 3:

**Figure supplement 1.** Classical trail.

**Figure supplement 2.** Distance of first mark in bout from load.

allows the group to avoid a deadlock (*Figure 5B*, left side). The inevitable cost of this is that as the load follows marks that are correctly directed to the right, it temporarily switches its direction against the useful advice (*Figure 5B*, right side). Taken together, the time saved near the slit outweighs the time lost on the non-productive detour and the load rapidly circumvents the obstacle. As predicted by our model, this motion is near-optimal in the sense that the mean passage time scales linearly with obstacle width (*Figure 5C*).

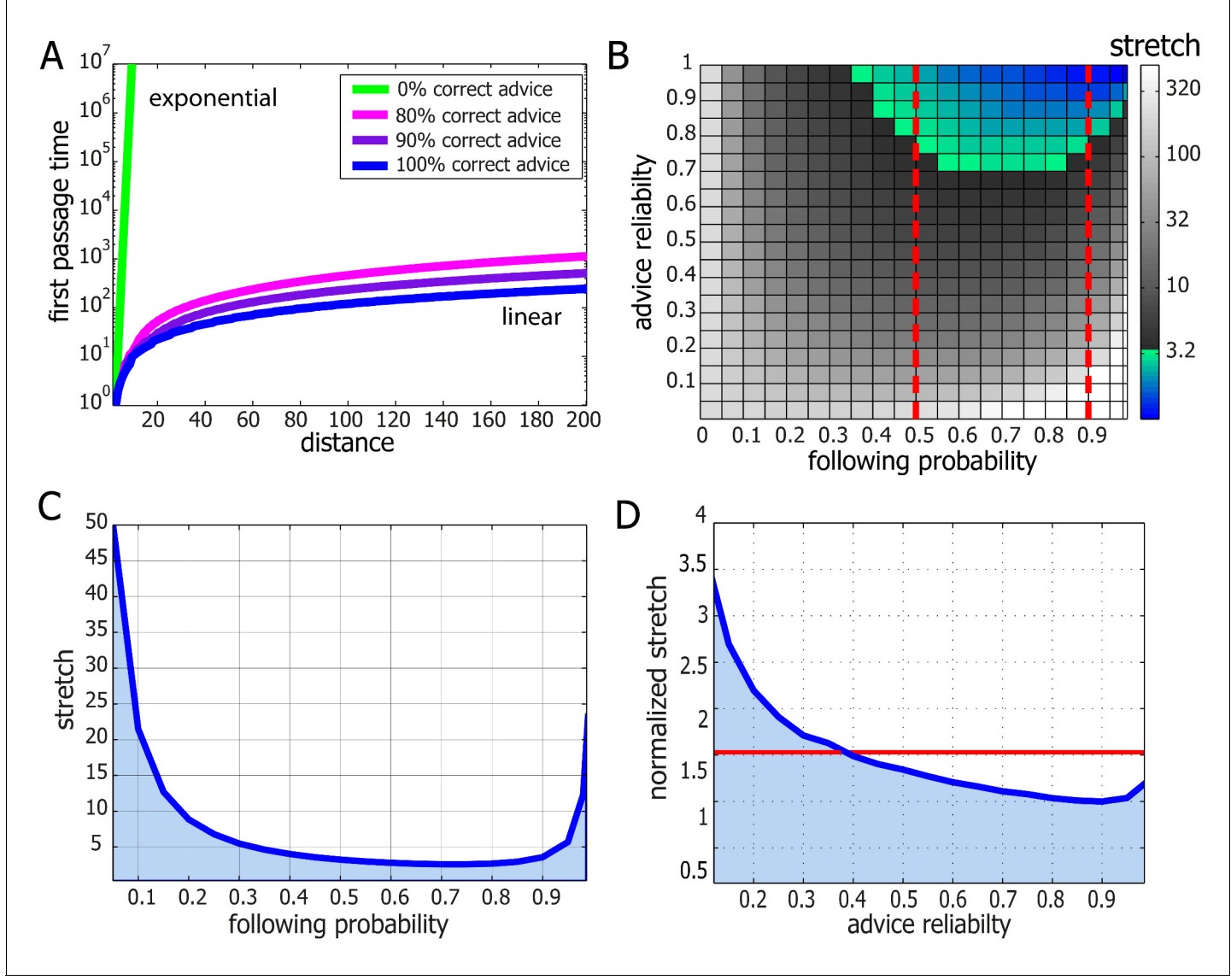

**Figure 4.** Performance and robustness of *Probabilistic Following* – simulation results. (**A**) Median first passage times of an agent performing *Probabilistic Following* on a line using a following parameter of 0.8. Different plots signify different reliability of advice. When advice is all or mostly correct (blue, purple, and magenta curves), the agent achieves fast, linear passage times (for a linear scale plot see *Figure 4—figure supplement 1*). When all advice points in the wrong direction passage times are exponential (linear green line in this logarithmic plot). (**B**) Stretch (see Materials and methods) of paths on a 2D grid for the full range of environmental advice reliability and the agent's probability of following the advice. The colored zone signifies stretch values that are under 3.2. Following probabilities between 0.5 and 0.9 (dashed red lines) are adequate for a large range of environmental values. (**C**) Stretch values for an advice reliability of 0.7 as a function of the agent's probability of following the advice. The local maxima near the edges indicate the poor performance of both a random walk (left-hand side) and a perfect following (right-hand side) strategy. The shallow minimum in between these indicates the lack of need for fine tuning to achieve near-optimal navigation times. (**D**) The stretch of an agent with a probability of 0.8 of following advice at a particular value of advice reliability (x-axis) normalized by the stretch of an agent which uses the optimal following probability for this particular environment. The value of one at an environmental reliability of 0.9 indicates that the agent with 0.8 following probability is the optimal at this environment. This same agent exhibits near-optimal performance (*i.e.* deviates from the optimum by less than a factor of 1.5, red line) for the full range of environmental reliability values over 0.4.

The following figure supplement is available for figure 4:

**Figure supplement 1.** Median first passage times of an agent performing *Probabilistic Following* on a line.

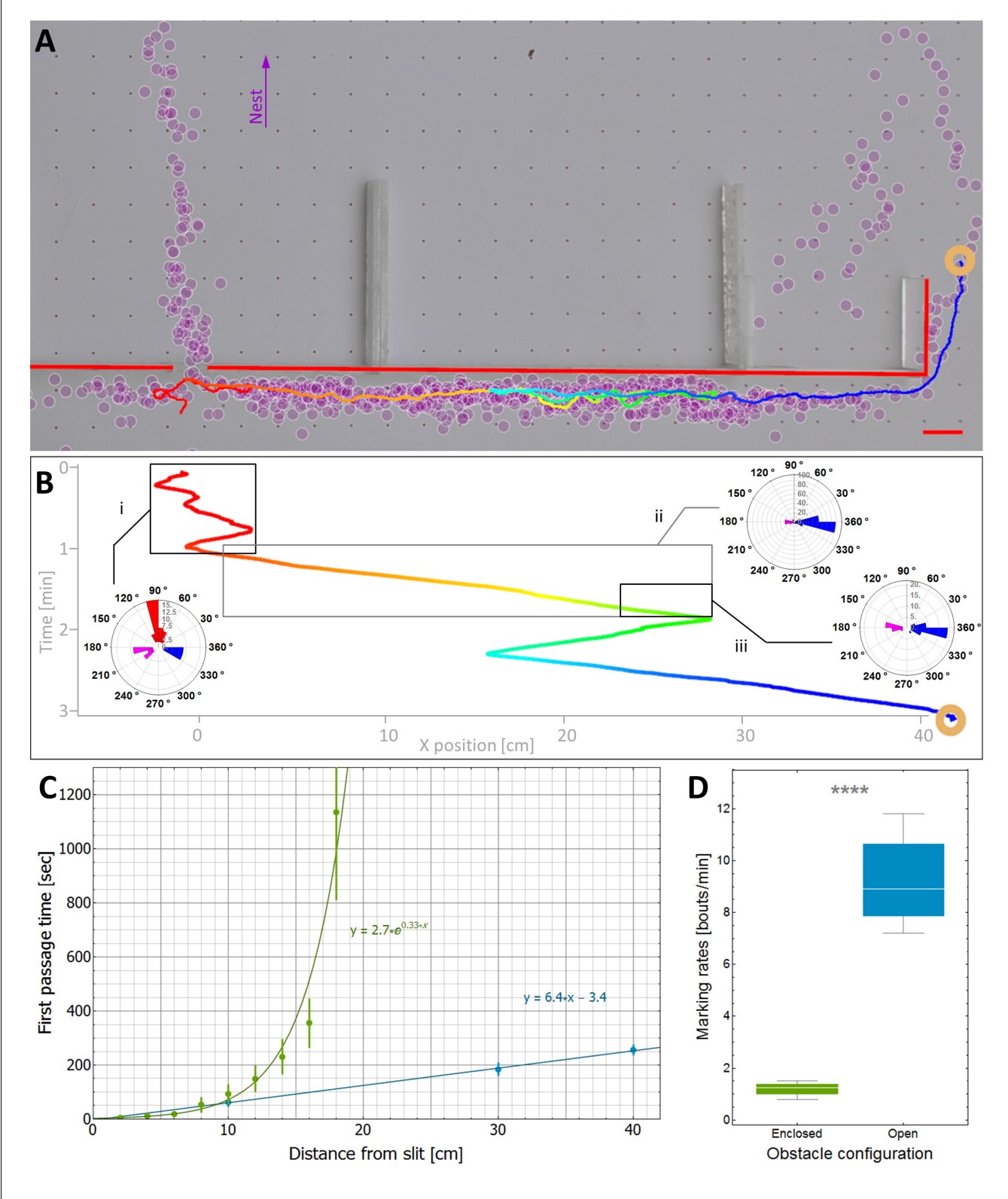

**Figure 5.** Efficiency and limitations of the locally-blazed trail. **(A-B)** A trajectory of a transported load (ring) while it moves around an 80 cm block with a slit at its center. Color code depicts temporal progression, these panels are spatially aligned along the horizontal axis. **(A)** 3205 marks (discs) identified during this transport, show a marking preference towards the nest (purple arrow) or along the walls. Short red scale bar denotes 2 cm. **(B)** X coordinate of the load as a function of time. Polar histograms present the scent marks density towards the nest (red), to the left (magenta) or to the right (blue),

*Figure 5 continued on next page*

*Figure 5 continued*

during three phases along the trajectory: (i) While the load was close to the slit, (ii) While moving to the right until the reversal point and (iii) in the last 5 cm just before the load reaches the same turning point. (C) The mean time to first reach a given distance from the slit along the obstacle walls is shown for both open- (Light blue) and enclosed-configurations (Green) with their corresponding best fit. The two obstacle configurations are illustrated in *Figure 5—figure supplement 1*. The log transformation of the data for the enclosed block, given in *Figure 5—figure supplement 2*, reveals exponential dynamics. (D) The marking rates along the obstacle walls are given for the two obstacle configurations. Asterisks denote p-value<$10^{-6}$.

The following figure supplements are available for figure 5:

**Figure supplement 1.** Two obstacle configurations.

**Figure supplement 2.** First passage time in collective motion directed against the scent marks.

## Further experimental validation

The previous section provided experimental support for the main prediction of our theoretical model. Here, we test further model predictions, both qualitative and quantitative, against the ants' cooperative transport behavior.

An important qualitative prediction of any model are the boundaries beyond which it is expected to fail. For *Probabilistic-Following* these are environments with large (compared to the persistence length of the collective motion) areas that contain mainly misleading information. As noted above, in such cases our theoretical model predicts slow traversal times that scale exponentially with the distance. To test this prediction, we challenged the carrying team with an 80 cm obstacle similar to that depicted in *Figure 5A* but modified such that the only path free for ant passage is through the slit and any bypass from the left or the right is blocked (see *enclosed-configuration Figure 5—figure supplement 1*). This confined obstacle is useful since it almost completely eliminates marking bouts in these perpendicular directions (see *Figure 5D*) such that, practically, all scent marks mislead the carrying ants towards the slit. We find that, in this case, in agreement with the model's predictions, the time it takes the group to travel away from the slit scales exponentially with distance (*Figure 5C* and *Figure 5—figure supplement 2*).

As predicted by the model, there is a stark contrast between linear traversal times when scent marks guide the ants around the obstacle and the exponential times when they guide it to the slit. This observation provides strong support for the importance of scent marks in the ants' obstacle circumvention behavior. This result has further consequences on the effect of direct interactions between the ants with the carried load and the obstacle. Such interactions are inevitable and include, at the least, the physical interaction between the load and the obstacle wall. While these interactions may induce some perpendicular motion (see *Figure 3C*) our result shows that, in general, this is insufficient to efficiently guide the group around the obstacle. Rather, scent marking and scent following are crucial components in this process.

The qualitative agreement between experiment and theory demonstrates the usefulness of introducing a general, intuitive, and analytically solvable abstract model. Although we do not claim that the collective motion can be fully described by our model, we provide quantitative evidence that it can serve as a good first approximation. We do this by comparing obstacle circumvention in the presence (*open configuration* depicted by blue *Figure 5—figure supplement 1*) or absence (*enclosed configuration* depicted by green in *Figure 5—figure supplement 1*) of useful scent marks. First, we estimate the rate at which the ants turn against pheromonal advice in the open-configuration experiment (see direction changes in *Figure 5B*) to be, on average, once per 37.5 cm (as measured over 15 trials and a total length of 636 cm). We can then use this rate to estimate the power of the exponential in *Figure 5C*. These two values are related since, in the enclosed-configuration experiment, all scent marks lead towards the slit and any movement away from it must result from instances in which the group moves against pheromonal markings. We find (see calculation in *Supplementary file 1-5*) that the correct relation between these values holds for a persistence length of 7 cm, in agreement with the 10 cm scale mentioned above, and an 80% probability of following advice, a value that lies well within the model's high efficiency zone (see *Figure 4B*).

# Discussion

To date, it was unknown whether ants employ scent marks during the retrieval phase of cooperative transport. Establishing an image analysis based method for the detection of scent laying events, we find that cooperative transport is indeed accompanied by significant pheromonal marking behavior (*Figure 1*).

The context in which these scent marks appear significantly differs from that of ants foraging on small food items, such as seeds, that are amenable for individual transport. A first difference is a direct consequence of the presence or absence of scale discrepancies between individual ants and the load they carry. During the transport of small items marking ants and carrying ants can, to a large extent, traverse the same paths (it is often the case that laden ants concurrently mark the path). In this case, the information communicated by trail laying ants is directly valuable to those that retrieve the food. This is not the case when large food items are involved. Simply put, in the context of cooperative transport, a passage that is open for a marking ant may be inaccessible for the larger load. Accordingly, we find that while scent marks typically point the group in the correct direction (*Figure 2A–C*) they may, at certain topological circumstances, be systematically misleading (*Figures 2D*, *3C* and *5A–B*). A second difference between the two contexts involves the degree of dynamics that characterizes the navigational challenge. Scent marks deposited during the transport of small food items assist traffic between two stationary locations (e.g. the pile of seeds and the nest) (*Franks et al., 1991*; *Bruce and Burd, 2012*; *Plowes et al., 2013*; *Bottinelli et al., 2015*; *Buhl et al., 2009*). Conversely, during cooperative transport, scent marks bridge between a fixed location (the nest) and a dynamically moving object (the carried load).

In light of these differences – we find that, during cooperative transport, ants employ a trail that is morphologically distinct from previous descriptions of ant trails. First, this trail is locally-blazed, *i.e.* it is composed of short broken segments that are dynamically laid in the near vicinity of the moving load. Unlike classical ant trails where individual ants lay scent marks over long distances that are on the length scale of the distance to the nest (*Cammaerts and Cammaerts, 1980*; *Beckers et al., 1992b*), here, marking ants indicate only the next step to be taken (*Figure 3D*). This highly transient role of the scent marks is reflected in their chemical makeup where the major component is the highly volatile pheromone undecane. Second, classical ant trails are typically clearly bounded and narrow. This is a result of the relatively strict manner in which ants follow the scent marks (*Bruce and Burd, 2012*; *Eidmann, 1927*; *Carthy, 1951*) which dominates over a secondary stochastic process where some ants distributively and individually wander off the trail (*Beekman and Dussutour, 2007*; *Deneubourg et al., 1983*; *Perna et al., 2012*; *Gordon, 1995*; *Goss et al., 1989*; *Reid et al., 2011*). The situation is very different for the locally-blazed trail which does not have well defined boundaries. This is a result of repeated occasions in which the entire group of carrying ants collectively and simultaneously loses the trail (*Figure 3B*). In turn, the trail is then reassembled from another location (*Figure 3A,C*).

It is an interesting question whether the unique characteristics of the locally-blazed ant trail are compatible with the navigational challenges that are specific to cooperative transport. A first challenge is the highly dynamic nature of the cooperative transport process wherein the constant motion of one of the trail's end-points (i.e. the food load) leads to low predictability. Above, we have discussed how the local nature of marking behavior is compatible with these circumstances. A second main challenge that must be overcome during cooperative transport is the high propensity for deadlocks. Here, the challenge faced by the ants is not only to overcome isolated deadlocks but, rather, to successfully go through a long sequence of decisions while avoiding the accumulations of errors that may rise due to maladaptive choices. Below, we discuss the efficiency of stochastic following in this context.

In general, it is theoretically established that stochastic components may sometimes grant search algorithms with the ability to escape from local minima (*Vermorel and Mohri, 2005*; *Selman et al., 1998*; *Kirkpatrick et al., 1983* ). In particular, it has been shown that this principle is utilized by ants on classic foraging trails. In this case, noisy trail following can resolve possible deadlocks that may occur when a high quality food source is introduced to ants that have already established a trail to a mediocre food source (*Beekman and Dussutour, 2007*; *Deneubourg et al., 1983*). In a similar manner, we predicted that the stochastic following behavior in the locally-blazed ant trail (*Figure 3B*)

plays an important role in resolving deadlocks. We then validated this prediction experimentally (*Figure 3C*).

While stochasticity can help groups to collectively escape from isolated deadlocks it is by no means a given that it is a sufficient component for an efficient navigational strategy which, in the context of cooperative transport, must employ a sequence of highly dependent decisions while overcoming misleading information. Indeed, the addition of noise cannot be considered as a silver-bullet that increases the efficiency of any search algorithm and there exist scenarios in which it simply does not work (*Selman et al., 1998*; *Reingold, 2005*; *Patrascu and Thorup, 2007*). Understanding how the locally-blazed trail allows ants to achieve near-optimal navigation performances (*Figure 4A–C*) is the subject of the theoretical aspects of this work.

We conducted an algorithmic study to obtain a more fundamental understanding regarding the efficiency of the locally-blazed trail as a navigational strategy. Specifically, we studied an ant-inspired algorithm which includes imperfect (stochastic) responsiveness to local, occasionally misleading, advice. An intuitive version of this problem involves driving a car in an unknown country that is in the aftermath of a major hurricane which has randomly flipped a certain small fraction of the road-signs. A driver's goal is to reach her destination as quickly as possible. If the driver chooses to follow all signs, she can be trapped in an infinite loop. Altogether ignoring signs is also inefficient since this fails to utilize the useful information that is contained in unflipped signs. We analyze the algorithm in which the driver follows any road-sign she encounters with some fixed probability and takes a random direction otherwise. Adapting results from the mathematical field of Random Walks in Random Environments (*Drewitz et al., 2014*; *Sznitman, 2002*), we have shown that this ant-inspired algorithm yields near-optimal navigation in topologies that are relevant to ant cooperative transport. Furthermore, the driver need not know exactly what fraction of the road-signs have been flipped. The robustness of the algorithm (see *Figure 4*) implies that the time it takes to reach the destination does not strongly depend on the probability at which the driver chooses to ignore the road-signs. The lack of a requirement for fine-tuning is a favorable trait when modeling biological systems which are inherently noisy and must function in contexts of unforeseen external challenges.

In the context of the intuitive example presented above, following each road-sign with some fixed probability works to help the driver reach her destination quickly, at least on 2D grids. We propose this mechanism as a simple strategy that may prove its efficiency in multiple contexts in which an agent needs to act in the presence of typically correct but, at places, systematically misleading advice. We note, however, that a rigorous proof would highly depend on the particular given model and may become extremely hard mathematically. In fact, even proving that this strategy holds with respect to navigation in general graph structures seems to be beyond the scope of currently known techniques (see *Supplementary file 1-4*), and remains a challenging mathematical question for future theoretical investigations.

Above we have compared the locally blazed trail to previously described ant trails. It is further interesting to discuss this form of collective navigation in the context of collective decision making in other animal groups. To remain as a cohesive group, social animals have to take decisions collectively (*Conradt et al., 2007*). A common strategy for reaching quality decisions is to follow the majority (*Ame et al., 2006*). Majority based averaging mechanisms have been shown to help the group reach accurate decisions even when its members' individual perception is inaccurate and noisy (*Dell'Ariccia et al., 2008*; *Simons, 2004*). Majority based decision making has also been demonstrated to be helpful even in the context of conflict (*Couzin et al., 2011*): It is often the case that information perceived by each individual group member encompasses only a small fragment of the complete environmental state. This may result in a case wherein different group members drive the group towards different, often disjoint, collective choices (*Franks et al., 2002*; *Seeley et al., 2012*; *Biro et al., 2006*). However, if individuals sense their environment in a reliable manner then favorable choices will be supported by a larger number of individuals than less favorable ones. In this case majority boosting techniques such as opinion polling (*Franks et al., 2002*; *Seeley et al., 2012*), quorum sensing (*Franks et al., 2002*; *Seeley et al., 2012*), and collective motion patterns (*Couzin et al., 2005*; *Strandburg-Peshkin et al., 2015*) can assure that the group as a whole followed the best alternative.

Cooperative transport through natural environments presents a different sort of challenge. While the majority opinion is usually reliable, at certain locations scale differences between an individual and the carried item can cause it to be plain wrong. Accordingly the locally-blazed trail includes two

different movement components. The first stands in agreement with the examples presented above: namely the group follows the majority opinion as evident in the scent mark pattern (see *Figure 5B*). The second component ignores the scent marks or majority opinion and moves in a different direction. Our model shows that randomly switching between these two behavioral patterns yields near-optimal navigation times.

In a dynamic environment, majority opinions may grow to be outdated (*Rieucau and Giraldeau, 2009, 2011*; *Warner, 1988*). Animal groups have been shown to produce innovation even when this contradicts the natural tendencies to conform to the majority (*Parrish, 1991*; *Reader and Laland, 2001*). The capacity for innovation has been observed in a diverse array of group living animals including fish (*Brown and Laland, 2002*), ants (*Beekman and Dussutour, 2007*; *Deneubourg et al., 1983*; *Czaczkes and Heinze, 2015*), mammals (*Reader and Laland, 2001, 2012*), and birds (*Rieucau and Giraldeau, 2011*) that rely on mechanisms such as preference of individual information to social information (*Rieucau and Giraldeau, 2011*), the addition of random components to individual behaviors (*Beekman and Dussutour, 2007*; *Deneubourg et al., 1983*), reduction of social signals (*Czaczkes and Heinze, 2015*), and deliberate individual exploration (*Brown and Laland, 2002*). It has been shown how this capacity allows the group to break away from suboptimal behavior (i.e. local minimum) (*Beekman and Dussutour, 2007*; *Deneubourg et al., 1983*; *Reader and Laland, 2001*; *Day et al., 2001*). The noisy component of the locally blazed ant trail falls within this group of collective phenomenon. An interesting distinction is that while previous observations often focused on noise and exploration in the context of a single choice we demonstrate its usefulness in a more complex situation involving a string of strongly dependent decisions. We show both theoretically and experimentally (*Figure 5*) how the locally blazed trail suffices to efficiently navigate the group over long distances. Our work suggests that the usefulness of noise and local exploration in animal groups extends beyond short time scales and single decisions and may actually be efficient for the long time scales and multiple decisions that characterize any biological system.

Finally, an ant colony is a hierarchical structure (*Wilson and Hölldobler, 1988*) that can be studied on various scales of organization. In this work, we focused on the scale of the team (*Anderson, 2001*) of carrying ants and studied the relations between its collective motion and the pheromone marking. It is, of course, interesting to understand how these relate to the single ant level. Although it is beyond the scope of this paper, below, we provide some hypotheses and directions for future research. One aspect of this concerns the mechanism by which individuals decide to lay scent marks. Such decisions may involve the ant's general knowledge of the environment (*Collett and Collett, 2002*), her recent history (*Wystrach et al., 2013*), and her interaction with obstacles or passageways. The differences in scent mark laying behavior according to the large-scale structure of an obstacle, as presented in *Figure 5D*, suggest an experimental scheme for testing the interplay between these factors. The decision when and where to lay scent marks may further be influenced by the presence of previously laid pheromones. Note however, that a positive feedback between marking and further marking is not required for the cohesiveness of the trail itself; Marks may be in close proximity simply because they all originate at the load and point in roughly the same direction (towards the nest). This may break down in the presence of obstacles where several alternative routes to the nest are available. Indeed, in this case, we observe that not all scent marks agree in their direction (see *Figure 5B*) such that no strict consensus is enforced. Importantly, such lack of consensus does not hold significant implications to our theoretical analysis (see *Supplementary file 1-1*).

It is further interesting to study how the relationship between load movement and the locally blazed trail emerge from the interactions of individual ants with individual pheromone marks. Previous work showed that, in *P. longicornis*, steering of the carried load is accomplished by the pull exerted by newly attached ants (*Gelblum et al., 2015*). We hypothesize that non-attached ants interact with the scent marks and that this affects their pulling direction once they attach to the load. Under this hypothesis, the response of the carrying group to the scent marks is not direct but, rather, mediated by these newly attached ants. Verifying this hypothesis is the subject of future work.

To summarize, cooperative transport is an experimentally tangible phenomenon that allowed us to probe the inherent conflicts between the different organizational scales. These conflicts lead to persisting errors in the navigational instructions that individual ants provide to the group. We have shown how random following behavior with respect to short marking trails works to resolve these conflicts and allow for efficient collective performance.

## Materials and methods

### Experimental setup

Data was collected from six ant nests in Rehovot, Israel. Tests were carried out, in the field, during the summer when *P. longicornis* ants display cooperative transport behavior. In general, a 100 cm × 70 cm laminated paper sheet was placed near the ants' nest on which either a Cheerio or a 1.5 mm thick ring-shaped piece of silicon (incubated over-night in either Royal canin, Aimargues, France, or Happycat, Wehringen, Germany, brands cat food) was introduced. In some trials obstacles were introduced between the location of the food and the nest entrance. Perspex obstacles of width 20, 60, or 80 cm were coated with Fluon (Sorpol) to prevent ants from climbing over them, and had a 5 mm slit at their center such that ants could easily pass but the object cannot. Side-view movies were filmed as the ants were transporting a load while walking on an elevated surface covered by millimeter paper. Two cameras, filming at the same frame rate, were placed above and to the side of the load's trajectory and were manually synchronized according to the ants locations on the millimeter paper. A scenario in which ants do not hold information about the direction to the nest was created by gently picking up the load together with the ants that were connected to it and laying it on a 40 cm × 30 cm clean paper sheet, with no freely moving ants. The lack of directional information was evident by the load's random walk motion rather than the ballistic motion observed without the manipulation (for details see *Gelblum et al., 2015*).

The data were filmed using either a Canon EOS 550D camera or a Panasonic HC-VX870K camcorder. A designated image processing program, developed using MATLAB programming language and described in our previous study (*Gelblum et al., 2015*), enabled high-quality tracking of the ants' location as well as the location and orientation of the transported load.

### Scent mark identification

We developed a method that allows us to pinpoint the pheromone marks laid by individual ants by using their velocity profile.

Typically, the speed signature of a marking event lasts about 0.2 s and involves a sharp deceleration followed by high acceleration phase (*Figure 1B and C* and *Video 3*). Candidate marking events were found by screening the speed profiles of the non-carrying ants with a time window of 0.25 s. The selection criteria included the existence of a local speed minimum lower than 3.5 cm/s and absolute value of deceleration/acceleration around the minimum that is greater than 30 cm/s$^2$ indicating a sharp stop-and-go episode. These criteria capture more than 97% of the true marking events as well as multiple false positives (non-marking episodes) whose percentage vary considerably due, for example, to variations in the number of non-marking ants present in the vicinity of the load.

To rid the dataset of these false hits, the automated stage was followed by a second manual stage. Here, suspect marking events automatically extracted from the speed profile, were further examined by a human observer who reviewed the corresponding raw video footage, where more subtle typical movements (e.g., slight distortion of the gaster shape, ant shadow location relative to the gaster, and the speed reversal described in *Figure 1C*) are visible (see close-up visualization in *Video 3*). This allowed differentiating true marking events from other stopping behavior showing similar velocity profiles.

To test the reliability of the entire marking detection procedure, we simultaneously filmed ants from both the top and side view (*Figure 1—figure supplement 2* and *Video 2*). Marking behavior was detected, as described above, from the top-view video and then verified against the corresponding side view video. We find that our method is highly reliable (Identification rates are 93% of true positives and a positive likelihood ratio of 6.75).

### Chemical analysis

Trail pheromone samples were collected in the ants' natural environment at the same site used for obstacle bypass assays. The pheromones were collected on thin layer chromatography (TLC) plates of silica modified with covalently bonded octadecyl on a glass support (Analtech, Newark, DE). The plates were precut to a size of 5 × 20 cm$^2$ and thoroughly cleaned using ethyl acetate, hexane and acetone. The plates were stored in a closed polypropylene box until arrival to the site. Pheromone samples were collected by placing a TLC plate between the nest and an immobilized object which

the ants attempted to carry to their nest. The ants were allowed to freely mark the plates for 5 min after which the plates were placed on dry ice to minimize pheromone evaporation. Control experiments were done by creating a scenario were ants walk on the TLC plates without marking it with trail pheromones. To do this, TLC plates were placed on poles such that ants could not reach them and they could not form a trail to this area. Ants were continuously collected from the floor and placed on the TLC plates for 10 min after which the plates were placed on dry ice. To prepare the samples for analysis, silica was scraped of the surface of the glass and immediately transferred to clean glass vials containing 3 ml hexane. The samples were sonicated for 20 min after which the supernatant was collected and transferred to new vials. Excess solvent was evaporated under a nitrogen stream to a total volume of 70 μl.

Samples were analyzed on a 7890 Agilent gas chromatograph coupled to time of flight mass spectrometer equipped with a Gerstel cooled injection inlet and a fused silica column (DB5-MS, 30 m × 0.25 mm, Agilent) and compared against a linear hydrocarbon mix standard. The large volume injection mode was used to increase sensitivity with an injected volume of 50 μl. Inlet temperature was set to $-21°C$, the vent flow was set to 260 mL/min at 7.5 PSI and the injection speed was set to 1.04 μl/sec. The inlet temperature was kept for 1 min after which it was heated to 260°C at 720 degrees/min. The oven program started at 30 where it stayed for 3 min after which it was raised to 310°C at eight degrees/min where it stayed for 10 min. The instrument was operated at constant flow of 1 ml/min.

## Calculation of scent marks information content

We estimated the amount of information that a single scent mark conveys about the direction between the load and the nest using 1395 scent marks recorded over several meters of collective motion, and gathered in several different occasions.

We considered each scent mark at the moment it appears. We measured the angle between the line that connects the center of the load to the nest and the line that connects the center of the load to the scent mark. The resulting collection of angles can be regarded as messages conveyed by the ants to the load regarding the direction to the nest. The uncertainty in these messages was quantified by grouping these measurements into 36 bins of 10° each (bottom panel of *Figure 2A*) and estimating the entropy of the resulting normalized probability histogram, pi: $H_{absolute} = -\sum p_i \cdot \log_2(p_i)$, where the sum is taken over all 36 bins. This uncertainty could be compared to the case in which no message was received such that any of the 36 possible directions is equally probable: $H_{no\,message} = -\sum \frac{1}{36} \cdot \log_2\left(\frac{1}{36}\right) = \log_2(36)$. The average information, in bits per mark, is estimated as the reduction of entropy between these two cases: $I_{absolute} = H_{no\,message} - H_{absolute}$.

We used the same dataset to estimate the amount of directional information that the load gains, on average, from a single mark. To do this, we compared two probability histograms. The first, $p_i^{before}$, is a histogram of the angles, $\theta_{before}$, between the line connecting the center of the load to the scent mark (at the time of marking) and the tangent to the load's trajectory in the two seconds that precede the appearance of the mark (bottom panel of *Figure 2B*). The second histogram, $p_i^{after}$, is similar but is calculated over angles, $\theta_{after}$, between the line connecting the center of the load to the scent mark and the tangent to the load's trajectory in the 2 s that follow the marking event (bottom panel of *Figure 2C*). While both of these histograms are centered around zero, the second one is narrower. This indicates that the object adjusts its direction of motion by turning towards the mark in the two seconds that follow its appearance. To quantify this reduction in histogram width we calculate the entropies of the distributions before and after the marking: $H_{before/after} = -\sum p_i^{before/after} \cdot \log_2\left(p_i^{before/after}\right)$. The amount of information, in average bits per mark, that is gained by the load from each such event was estimated by the reduction in entropy in the seconds that follow the marking event: $I_{load} = H_{before} - H_{after}$.

## Marking bout measurements

To measure the distance of the first mark in a bout from the object we used the video footage to identify N = 735 marking bouts and measure the distance of the first mark from the center of the load. We observed that marking bout distances can vary between several meters (*e.g.* for the recruiting ant) and several centimeters (*Figure 3D*). To measure the distribution of bout lengths, we used a Disto-D3 distance meter (Leica Geosystems; Heerbrugg, Switzerland). Ants were induced to carry an

object at distances of 5–7 m from the nest during which we identified ants that commence marking near the object (this is justified by our measurement indicating that marking bouts commence in the close proximity of the carried load *Figure 3—figure supplement 2*,). A person holding the distance meter marked the location of the object at the beginning of the bout using a small marker while a second person tracked the marking ant. Once the ant marks the furthest mark from the load, a distance measurement was taken. Note that using video techniques to track fast moving 3 mm ants over many meters in the field is impractical.

## Numerical simulations

Simulation of routing in grids and lines was performed under the following conditions. The network was obtained by placing N uniformly-spaced nodes in the form of a grid of dimensions $\sqrt{N} \times \sqrt{N}$, or in the form of a line of length N, respectively. For the studied graphs, we considered an unreliable advice model in which each node with probability p had its advice chosen to point along a shortest path towards the target (with ties between different possible shortest paths on the grid broken so that the selected paths towards the target always formed a tree structure), whereas with probability (1-p) this advice was chosen to point to a neighboring node, chosen uniformly at random. The routing process was subsequently performed without any updates to the advice. A routing protocol which performs shortest path routing solely by following the advice would therefore be likely to be stuck at some intermediary node of the considered environment.

All simulation results were obtained through a Monte Carlo simulation. For the grid, we tested the performance of the PF algorithm for different values of probability of following advice in the range [0,1] and a fixed number of nodes N = 50 × 50 of the network. Each data point corresponds to a random sample of 1000 graphs under the studied model of unreliable advice, in which the routing process was repeated 50 times for different source nodes, assuming that a node located next to the center of the grid was the target. For the line, we varied the number of nodes in the range N ∈[2 200], starting from the leftmost node and routing towards the rightmost node. For each length of the line, we sampled between 104 and 106 advice configurations under the studied advice model.

The quality of a specific execution of the *PF* routing process between a source-target pair in a graph is represented by its stretch, defined as the ratio between the number of hops traversed when performing PF routing and the number of hops of the actual shortest source-target path in the updated graph. For the grid, the aggregated parameter presented in *Figure 4B–C*, called effective stretch, represents the average value of stretch in each scenario, taken over 90% of the obtained data-set after discarding executions with the largest stretch. Such a definition of effective stretch ensures the meaningfulness of simulation results, considering that the expected value of the studied random variables may, in some settings, be potentially unbounded. In *Figure 4D*, this effective stretch parameter is subsequently normalized with respect to the effective stretch of the optimal following probability. For the line, the parameter presented in *Figure 4A* is the median stretch over all runs.

## Statistical analyses

Box plots representations (*Figure 1D*) denote the median values inter-quartile range (box) and the lowest datum still within 1.5 times the inter-quartile range (whiskers).

Error bars for all information content measurements were estimated using a bootstrap method (technical replication). For each non-normalized histogram, we generated 10,000 noisy histograms in which each bin *i* takes the value $q_i'=q_i + \beta_i \cdot \sqrt{q_i}$ where $q_i$ is the value of the bin in the corresponding original histogram and $\beta_i$ is a random noise term with values in the interval *[1, 1]*. These histograms were then normalized to obtain probability histograms ($p_i$) from which were used to generate a distribution of entropies (using the exact same procedure as described in the 'Calculation of scent marks information content' section). This procedure ensures that bins with more measurements are associated with smaller errors in the estimation of the corresponding probabilities. The standard deviations of these distributions were taken as the errors of the entropy measurements, and thus of the information.

Error bars in *Figure 3C* signify the SEM (Standard error of the mean) using, for each point, a sample size that equals the number of measurements it includes and assuming, again for each bin, a binomial distribution with a success probability corresponding to the y-axis measurement. The SEM

is estimated as the standard deviation of this for this binomial distribution divided by the square root of the sample size.

The probability that the marking bout distances of the first recruiter ant can be considered as samples of the distribution characterizing the locally-blazed ant trail as depicted in *Figure 3D* was calculated by the probability of obtaining twelve independent samples of over six meters from this distribution. The probability of obtaining a single such sample is the number of measurements of six meters and over the total number of measurements which is $p = 16/735 \approx 0.022$. The probability of twelve consecutive such measurements is $p^{12} < 10^{-6}$.

## Acknowledgements

We thank Gershon Elazar, Guy Han, and Yosef Shopen for technical help. This research is supported by the European Research Council (ERC) under the European Unions Horizon 2020 research and innovation programme (grant agreement No 648032). E.F. is the incumbent of the Tom Beck Research Fellow Chair in the Physics of Complex Systems. A.Kos. was partially supported by ANR (France) Project DESCARTES and by NCN (Poland) grant 2015/17/B/ST6/01897. A.Kor. was partially supported by the ANR projects DISPLEXITY and PROSE, and by the INRIA project GANG. O.F. is the incumbent of the Shloimo and Michla Tomarin Career Development Chair, was supported by the Israeli Science Foundation grant 833/15 and would like to thank the Clore Foundation for their ongoing generosity.

## Additional information

### Funding

| Funder | Grant reference number | Author |
| --- | --- | --- |
| European Research Council | DBA-648032 | Amos Korman<br>Ofer Feinerman |
| Israel Science Foundation | 833/15 | Ofer Feinerman |
| National Science Centre | 2015/17/B/ST6/01897 | Adrian Kosowski |

The funders had no role in study design, data collection and interpretation, or the decision to submit the work for publication.

### Author contributions

EF, AKor, OF, Conception and design, Acquisition of data, Analysis and interpretation of data, Drafting or revising the article; YH, LB, AKos, Acquisition of data, Analysis and interpretation of data, Drafting or revising the article; AG, Analysis and interpretation of data, Drafting or revising the article

### Author ORCIDs

Ofer Feinerman, http://orcid.org/0000-0003-4145-0238

## Additional files

### Supplementary files

• Supplementary file 1. Definitions, proofs, and calculations pertaining to the theoretical model.

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
