## [Decision Letter]

Thank you for submitting your article "A Locally-Blazed Ant Trail Achieves Efficient Collective Navigation Despite Limited Information" for consideration by *eLife*. Your article has been reviewed by two peer reviewers, one of whom is a member of our Board of Reviewing Editors, and the evaluation has been overseen by Eve Marder as the Senior Editor. The reviewers have opted to remain anonymous.

The reviewers have discussed the reviews with one another and the Reviewing Editor has drafted this decision to help you prepare a revised submission.

The authors describe navigational decisions of *Paratrechina longicornis* ants cooperatively transporting a large food item. In contrast to other species and this species in other circumstances, these animals use a locally-blazed trail, marking short segments beginning at the site of the load. Subsequently, animals may lose the guiding trail but successfully regain their direction using partially available information. In doing this experiment, the authors' used side view cameras allowing detection of putative scent marking activity over a large two-dimensional surface.

The reviewers had some concerns that need to be addressed in the text:

1) There is some concern that the authors did not experimentally test or validate a prediction from the model. And that in the Discussion, other examples of animal cooperation (e.g., birds, bats wolves, etc.) could be compared and contrasted.

2) Each ant makes its own decision, independently of other ants, so the problem seems to require additional mechanisms to keep the individual's cohesive with the group. How is this accomplished? How does it affect the PF algorithm, and its performance? Does the algorithm require a leader?

3) Are there any biological predictions that can be made from the model? For example, values of model parameters, or their relationships? Theorems suggest a very precise relationship between some of these parameters (λ, mu, q, etc.) in order for the algorithm to work. Does this relationship also hold true for ants? While it is always difficult to relate complex biological parameters to abstract model parameters, this would at least provide some feedback from the model back to learning new biology. Moreover, the model assumes parameters values are fixed, but the data seem to suggests that ants likely change their behavior when an obstacle is reached, preferring to explore more. Isn't this an important biological factor to consider?

4) Third, it's not clear what the role of pheromone is in this model when an ant decides its next move. For real ants, it seems like pheromone may be important and is what contributes to the fact that the classical trail is stable over a long time, whereas the locally-blazed trails are less stable. How does this interface with advice (which seems like a distance-dependent marker, and not a typical pheromone marker indicating previous activity)? Are there possibly different pheromones?

5) Fourth, the analysis of MANETs demonstrates a real-world application, but is this necessary? If this is to be included, the authors need to compare their algorithm with state-of-the-art MANET routing algorithms (e.g. gossip algorithms). Figure 4 shows a stretch of 10-50x, which seems quite bad to me, but maybe it isn't. Without a comparison, it is difficult to appreciate Figure 4, and it doesn't provide any evidence that the probabilistic algorithm the authors suggest is valuable to this well-established community.

---

## [Author Response]

[…]

*The authors describe navigational decisions of Paratrechina longicornis ants cooperatively transporting a large food item. In contrast to other species and this species in other circumstances, these animals use a locally-blazed trail, marking short segments beginning at the site of the load. Subsequently, animals may lose the guiding trail but successfully regain their direction using partially available information. In doing this experiment, the authors' used side view cameras allowing detection of putative scent marking activity over a large two-dimensional surface.*

This is not precise, what allowed us to monitor marking activity over large two-dimensional surfaces is the use of a top-view camera. This ability was the main technological advancement that differentiates our work from previous works (cited in our manuscript) that used side view cameras to identify scent marks but could do this only over narrow one-dimensional strips. In our work, side view cameras were only used in the verification and accuracy evaluation stages of the top-view based method.

*The reviewers had some concerns that need to be addressed in the text:*

*1) There is some concern that the authors did not experimentally test or validate a prediction from the model.*

Our paper presents the locally-blazed ant trail which appears during cooperative transport and significantly differs from all previously described ant trails. Intriguingly, the trail is characterized by traits that at first glance appear to be counterproductive to the ants’ navigational effort. These include the repeated instances in which the group completely loses what may be a perfectly good trail and the very short local and segmented pheromonal marking. The purpose of the model we present is not to accurately describe the ants' navigational mechanisms but, rather, to capture the governing principles of the locally-blazed trail and highlight the computational advantages of its counter-intuitive traits. As such, the model makes some general qualitative predictions regarding the locally blazed trail which we have experimentally tested, these are predictions A-B listed below. We further use new measurements to quantitatively test a new model prediction (see prediction C below). Results that have been added to the revision but were not included in the original submission are referred to below using the word "new".

A) Prediction: When moving with a typical speed, the travel time between two points can never scale faster than the distance between these two points. The single most important prediction of our model is that the locally-blazed trail will enable the ants linear scaling of travel times even in the presence of misleading information. This qualitative prediction of the abstract and therefore general probabilistic following model provides intuition to the counterintuitive aspects of the ants’ trail laying as outlined above. The linear time prediction sets a clear, qualitative and therefore major difference between this model and other possible navigational schemes: while a random walk model would yield an inferior, quadratic relation between distance and time and a model of perfect following would yield divergent time, our model predicts that the ants exhibit linear, near-optimal scaling (see also the new Figure 4).

Experimental Test:We measured the local behavior and global performance of the ant carrying group in the vicinity of an obstacle which induces areas of misleading scent-marks (see Figure 5). We extracted all scent marks in the vicinity of the load and analyzed the load’s motion.

Results:

First, this setting validates the main model assumption that the group motion follows the direction encoded in the scent-marks but not always. To test that this assumption indeed holds we extracted the scent marks laid by the ants during obstacle circumvention and compared them to the movement of the group. We indeed showed that while the group generally follows the direction suggested by the scent marks it sometimes switches direction and moves in the opposite direction (5B and the new data presented in the last paragraph of the section titled “Further experimental validation”). This strengthens our understanding that the assumptions of the model we describe indeed hold near obstacles with this topology.

By testing obstacles of varying size we find that the time to circumvent an obstacle with misleading scent marks is linear in the length of the obstacle (see Figure 5 and paragraph 3 of the section titled “Optimality of the locally-blazed trail”). We thus validate this most important prediction of our model.

These predictions and validations have appeared in the first version of the paper but it was not stressed that these are tests to the model’s main prediction. We make this point clearer in the current version of the manuscript (see the beginning of paragraph 2 and the end of paragraph 3 of the section “Optimality of the locally-blazed trail”).

B) New Prediction: An important aspect of any algorithm are the cases for which it provides a good solution and those for which it is expected to fail. Our model predicts when the ants' navigational strategy would break down. In fact, the time it should take the transported load to travel across large (compared to the 10cm scale that is mentioned in several points in this manuscript) areas that contain *only* misleading information would be very long and scale exponentially with the distance. This prediction is now presented in the last paragraph of the section titled “Fault-tolerant routing” which refers to the theoretical lemma 8 in [Supplementary-material SD1-data] as well as the new simulation results as presented in the now added Figure 4. It is mentioned again in paragraph 2 of the new section titled “Further experimental validation”.

New Experimental Test: We have constructed a modified obstacle (see Figure 5—figure supplement 1) that is, in every way similar to those used in the experiments described in Figure 5 (and in point A above), except that it prevents the ants from laying scent marks which point in the correct direction around the obstacle. Rather all scent marks lead towards the slit at the center of this 80cm obstacle. Note that the 40cm at each side of the slit allow for a large enough area (compared to 10cm) with misleading information only (as required by the prediction).

New Results: We verify that indeed there is a negligible amount of scent marks perpendicular to the direction to the nest (see the Figure 5). Therefore, the advice points towards the slit at every point. In these circumstances, the carrying group has difficulties escaping the effect of the slit. We show that the time it takes the ants to first reach a distance of *L* along the obstacle grows exponentially with *L* (see Figure 5 and Figure 5—figure supplement 2). Note that, in this context, other models for navigation would yield different results. For example, random exploration would result in passage time that scales quadratically with distance. These results are now presented in the 2^nd^ paragraph of the new section titled “Further experimental validation”. Note that this prediction further implies that the ant team's ability to apply a different navigational protocol when stuck at an obstacle is limited (3^rd^ paragraph of this section). This is further discussed in our answer to question 3 below.

C) New prediction: In our current version of the model we point to the robustness of our model. Mainly the degree of randomness which characterizes the ants’ motion does not have to be well tuned. In the second paragraph of the new section “Robust routing”, we now stress this theoretical result (which appears in [Supplementary-material SD1-data]). Furthermore, new results from our simulations show that a large area in parameter space at which our algorithm is expected to be efficient (see the new panels B-D of Figure 4). Quantitatively, this area includes following probabilities of between 50% and 90%.

Experimental Test: We tested for a quantitative match between two modes of the ants' collective motion: moving along scent marks (as in A) and moving against scent marks (as in B). This analysis uses two new measurements that were not included in the previous version of the manuscript: the rate of turning (in turns per cm) when going with the direction of the advice and the power of the exponential that describes the first passage time as a function of the distance when moving against the advice.

New Result: This comparison yields two values: a 7cm persistence length for the ant collective motion (this is in agreement with the 10cm expected scale) and a "following probability" of 80% (more accurately, the ant-team goes against the scent mark direction on about 20% of the time). This value lies well within the robustness zones predicted by our model as depicted in Figure 4. These results are now presented in the last paragraph of the new section titled "Further experimental validation" as supported by the newly added section 5 of the [Supplementary-material SD1-data].

*And that in the Discussion, other examples of animal cooperation (e.g., birds, bats wolves, etc.) could be compared and contrasted.*

The previous version of the paper included extensive discussion containing comparisons and contrasts between the locally-blazed trail and the more classical ant trail (see paragraphs 2-5 of the Discussion).

We have now added three paragraphs (paragraphs 9-11 of the Discussion) that relate our observation to collective decision making in other animal groups including cited examples regarding fish, birds, mammals, and other ant species. Mainly we discuss majority based decisions in these animal groups and the mechanisms by which majority rules must be ‘bent’ to allow for innovation.

We further discuss some general lessons that could be taken from the ant system to decision making in other animal groups: One (paragraph 11 of the Discussion) is that injecting noise into the decision making process could often prove as an efficient strategy for long sequences of decisions and not just for an isolated one as was usually considered. This, of course, depends on theoretically analyzing the situation at hand. The second (paragraph 7 of the Discussion) is that the noise in these decisions need not be fine-tuned. This relaxes the need that the animal group be constantly aware of changing environmental conditions.

*2) Each ant makes its own decision, independently of other ants, so the problem seems to require additional mechanisms to keep the individual's cohesive with the group. How is this accomplished?*

First of all, we wish to stress that the scale chosen in this study is that of the ant team and the pheromone trail (please see more on choice of scales in the answer to question 4 below). Conversely, this work does not focus on the scale of individual ants and we have not tried to decipher the decision making process which leads an ant to lay one or more scent marks in a particular direction. This is the subject of future work. This point is clarified in paragraph 12 of the Discussion (three paragraphs before the last).

That being said, we turn to answer the concerns you raise:

While the decisions made by individual ants may indeed be independent of each other two main factors contribute to the cohesiveness of the trail. First, the markings bouts performed by different ants all start from the object itself (see, for example, Figure 4). This, together with the locality of the markings, provide a mechanism which guarantees that most scent marks are laid in close proximity of each other. Second, ants that are unattached from the carried object are well informed of the direction to the nest. This can be seen, for example, by the recruitment trail of the first ant that finds the food bait whose trajectory back towards the nest is very well aimed. It can also be inferred from Figure 2 which exhibits a tight spread of angle of marks around the direction to the nest. This is despite the fact that markings were collected from different distant points along the trail (and are hence inherently independent). Therefore, in the case in which a single shortest path (from the perspective of individual ants) leads from the carried load to the nest, the marks will typically point to it. This, along with the shared bout starting point near the carried load, keeps cohesiveness between marking ants.

Near specific points in the vicinity of obstacles this picture may change. These are points where several passages to the nest are possible and, moreover, these passages are similar enough such that ants may differ in their knowledge of the preferred path. In these cases, it appears that the pheromone marks laid by the ants do not all agree and some minority “opinions” may exist (this is demonstrated in the polar plots of marking directionality as depicted in Figure 5). This provides evidence against the existence of a strict mechanism that works to ensure a consensus of marking to point towards a single direction.

To summarize, while markings are typically in agreement with each other there is no evidence for strong dependence between the different ants. This lack of dependence is exemplified by the fact that, in certain scenarios, the marks around the carried object may point in more than one direction and sometimes even to the opposite direction.

These points are now explained in paragraph 12 of the Discussion.

*How does it affect the PF algorithm, and its performance?*

The fact that the scent markings at a specific location do not all point in the same direction and that a minority direction sometimes exists does not bear a strong effect on the performance of the PF model or on the lessons we learn from it. When there are multiple pointers at a location the right way to model the fact that ants are typically correct but not always, would be to require that the probability that the strongest advice pointer points to a shortest path is at least some p. This provides a direct translation from this model to the Noisy-Advice model as the strongest directional mark is taken to be the advice at the location, and when running PF, the load only takes into account these pointers. Although it could be the case that all pointers at a given point are taken into consideration, we view the model that considers only the strongest pointer, as a first order approximation, which already demonstrates the basic properties of the navigation mechanism. Constructing a theoretical model that takes into account all pointers and balances between their strengths would be possible, but would complicate the presentation considerably, and we further do not believe that it will reveal a new insight that PF doesn't already have. This issue is now mentioned remark 1 of [Supplementary-material SD1-data] and referred from paragraph 12 of the Discussion.

*Does the algorithm require a leader?*

This question is not completely clear to us. Interpreting it along the main lines of question 2 we understand that the question is whether consensus in marking directions in a specific point is achieved via a leader ant that strongly influences the direction of marking. Since we do not always see consensus in markings (see Figure 5) then there is no evidence for such leadership in trail laying.

A second way of interpreting this question (somewhat disjoint from the subject of question 2) is whether any form of leadership appears during the cooperative transport process. As discussed in a previous paper of ours (Gelblum et al., Nat Comm 2015), the mechanical aspects of cooperative transport do rely on transient leadership. This issue is further discussed in our answer to question 4 below.

*3) Are there any biological predictions that can be made from the model?*

The predictions of the model are listed in the answer to question 1 above. These include the main qualitative prediction of linear passage time, a prediction of a situation in which the model breaks down and stops being efficient, and one more quantitative prediction regarding actual randomness levels. The robustness of the model to parameter tuning also has important biological implications as detailed in the answers to the subquestions of question 3 below. Further lessons to biology are listed in the answer to the second part question 1 above.

*For example, values of model parameters, or their relationships? Theorems suggest a very precise relationship between some of these parameters (λ, mu, q, etc.) in order for the algorithm to work.*

In fact, fine-tuning is not required for probabilistic following. Our theoretical results do indeed specify how the degree of randomness in trail following can be optimally tuned to environmental statistics. However, as we now stress detuning the following parameter away from its optimal value only marginally degrades the algorithm’s performance. In fact, wide ranges of the algorithm’s random following parameter yield near optimal behavior over a wide range of possible environments. We show this both theoretically: i.e., the wide range of the following parameter λ, as specified in theorem 13 of the Suplementary file 1, that still allows for linear performance, and by the simulations, presented in the new Figure 4, which quantitatively demonstrate this point.

We see this increased robustness as a very important trait of this model: it implies that the ants do not have to be fine-tuned to the statistical characteristics of their environment (which may change from day to day by, for example, the falling of leaves/branches creates new obstacles and changes environmental statistics). Rather, a large range of collective following parameters can be expected to function well over a large number of possible environmental conditions (see Figure 4). Along with its simplicity we see this as the second major strength of our model. As such we now thoroughly discuss the notion of robustness and mention its biological appeal in the newly added section titled "Robust routing" (supported by theorem 13 of the [Supplementary-material SD1-data] and the new simulations presented in Figure 4). The intuitive meaning and biological importance of robustness are now further discussed in paragraph 7 of the Discussion. Furthermore, due the importance of this point we now mention it in the Abstract as well. We thank the referees for pointing us in this direction.

*Does this relationship also hold true for ants? While it is always difficult to relate complex biological parameters to abstract model parameters, this would at least provide some feedback from the model back to learning new biology.*

We have now added new statistics and data to accompany the experiments described in Figure 4 that estimates the value of the ants following probability (λ). We show that this value falls well inside the range of near-optimal parameters that are predicted by our theoretical model. See prediction C in question 1 above, the fourth paragraph of the newly added section titled "Further experimental validation", as well as the newly added section 5 of the [Supplementary-material SD1-data].

*Moreover, the model assumes parameters values are fixed, but the data seem to suggests that ants likely change their behavior when an obstacle is reached, preferring to explore more. Isn't this an important biological factor to consider?*

When talking of “ants preferring to explore more”, one must differentiate between the unattached marking ants and the attached carrying ants.

Let's first discuss the marking ants. First, recall that this is not the scale of description on which this work is focused (which, rather, is that of the load and the scent trail). More on this in our answer to question 4 below. That being said, we still observe that near an obstacle individual ants may tend to explore more especially if the obstacle blocks their own path. As seen in Figure 5 near the obstacle marking ants tend to mark in a direction that is perpendicular to the original which will not happen as frequently in an open space (see Figure 2). Indeed, these markings are what guide the carrying group getting around the obstacle (see Figure 5). Moreover, the interactions of the individual ants with the obstacle are not local: Figure 5 show that whether an ant marks in a particular direction or not has to do with the structure of the obstacle far away from this point. We now explain and mention this in paragraph 12 of the Discussion where studying individual ant marking behavior is presented as a possible extension of this work (with experiments similar to that presented in Figure 5 serving as a good starting point).

Another question is whether the carrying group changes its behavior near an obstacle. The immediate answer is, of course, yes. The mechanical interaction with the obstacle physically blocks the previous direction of motion such that the collective behavior is forced to change. As the referees point out – the question is whether this is an important biological factor to consider.

The newly added Figure 5 and its accompanying panels describe a situation in which we challenged the ants with a modified block (the “enclosed-block”) that is identical in structure to the blocks as in Figure 5 but different from these does in that it does not allow for ant passage from the sides. We show that, in the vicinity of the group, the difference between the original obstacle and the modified obstacle is the absence of (correct) scent marks leading perpendicularly away from the slit in the modified obstacle (see the new Figure 5). Thus, this setup allows us to quantify the biological importance of direct interaction with the obstacle relative to that of the locally blazed trail.

We find that in the absence of correct scent marks (leading to at least one of the sides of the obstacle), the time it takes to first reach a certain distance from the slit scales exponentially with this distance (see Figure 5 – please compare to the prediction of Figure 4). This time reduces to linear in the presence of correct scent marks (see Figure 5 – please compare to the prediction of Figure 4). This implies that while direct interactions with the obstacle may have some minor positive effect (see Figure 3) they are by no means the cause of the fast linear obstacle circumvention times. This supports the claim that the efficiency of the ants’ motion can be attributed to the locally blazed trail. Therefore, without the scent marks, the interaction of the obstacle does not have a major biological significance. This point is now discussed in the 3^rd^ paragraph of the newly added section titled "Further experimental validation".

*4) Third, it's not clear what the role of pheromone is in this model when an ant decides its next move.*

When studying a biological system, one has to make inevitable choices regarding the scale of the observations. In this study, we have decided to focus on a scale that is larger than that of the individual ant. Hence, rather, instead of specifying the role of pheromone in how an individual ant decides her next move (as in your question) we study the role of pheromone in how the team of ants that cooperatively carry the large load makes its next move. Indeed, we show how the team of ants generally follows the scent marks but does not do exclusively this. We further show how this mode of navigation is valuable for efficient navigation of the scale on the whole team which is the length scale that is relevant for retrieving the large food item to the nest.

We do agree with the referees that it is intriguing how this rules are implemented on the "microscopic" level of single ants both attached and non-attached to the load. Understanding this is one of our next research goals. To conduct this analysis one must measure the trajectories of all individual ants moving around the object, relate them with recently laid scent marks, and then quantify the effect of the ants once they attach to the load and join the carrying team. This complex study is beyond the scope of the current paper and, as specified in the previous paragraph, not required for the Results and Discussion as presented.

The newly added paragraph 12-13 of the Discussion specifically states the organizational scale on which this study in centered and paragraph 13 (one before last) presents some hypotheses of how we believe individual ants react to the scent marks and how these individual reactions works to affect the moving direction of the load. The paragraphs state that testing these hypotheses is the subject of future work.

For real ants, it seems like pheromone may be important and is what contributes to the fact that the classical trail is stable over a long time, whereas the locally-blazed trails are less stable. How does this interface with advice (which seems like a distance-dependent marker, and not a typical pheromone marker indicating previous activity)? Are there possibly different pheromones?

In the manuscript, we show that the effect of pheromones is limited in distance. This is probably the result of two factors: fast evaporation and the probabilistic nature of load team's motion.

Indeed, it has previously been shown that *P. longicornis* ants lay trail pheromones with low boiling temperatures that display very fast evaporation (few minutes) compared to their own other trail pheromones lasting more than 24 hours as well as trail pheromones in other ant species that can endure for several days (and even up to 9 weeks without being reinforced. By using direct chemical measurements (which are, by the way, difficult and rarely performed) from the locally blazed trail we identified undecane (C_11_H_24_) as a main component, as was found in Witte et al., 2007. This is a short hydrocarbon molecule whose fast evaporation makes it a prevalent ant alert pheromone rather than a long-lived trail pheromone. We now provide this information in the first paragraph of the section titled "Identification of pheromone deposition events", refer to it again in the first paragraph of the section titled "The locally blazed ant trail" where we also mention the short-lasting responsiveness of the ants to the locally-blazed trail. The interface between local advice (both in space and time) and the volatile nature of undecane is now also noted in the third paragraph of the Discussion.

Finally, the probabilistic fashion in which the group follows the trail further contributes to the short range effect of the pheromone marking. This happens simply because pheromones that are laid far away from the load will not be reached by the team as it will lose is way earlier (see Figure 3).

*5) Fourth, the analysis of MANETs demonstrates a real-world application, but is this necessary?*

We agree with the referees that this technological application is certainly not the main point of our paper. We do, however, opt to keep a simulation study while presenting it under a different lens. This also changes the position in the manuscript at which these results are presented from the “Generalizations” section at the end (in the previous version) to the theoretical discussion in the new section titled "Robust routing".

We strengthen these simulation results by complimenting them with theoretical proofs (see theorem 13 of the [Supplementary-material SD1-data]). These proofs show that *PF* yields linear travel times for a wide range of the following parameter tuning. This is now emphasized in this newly added section as well as Figure 4.

As mentioned above, an important result which emerges from our theoretical study and the routing simulations is the high robustness of the proposed navigational model. We have added a new Figure 4 to demonstrate this. This provides a lesson for the biological world – namely that there is no real incentive for the biological system to measure the statistics of its environment or fine tune its motion parameters accordingly. Rather, using a very simple algorithm such as PF where the following probability falls somewhere within a wide range allows for very good performance within a wide variety of environmental conditions.

Further, these simulations allow us to make some quantitative predictions regarding reasonable values for the following probabilities of *Probabilistic Following*. Indeed we show that the values we observe experimentally fall well within this range (see prediction C in the answer to question 1 above).

*If this is to be included, the authors need to compare their algorithm with state-of-the-art MANET routing algorithms (e.g. gossip algorithms). Without a comparison, it is difficult to appreciate Figure 4, and it doesn't provide any evidence that the probabilistic algorithm the authors suggest is valuable to this well-established community.*

In accordance with the referees’ suggestion we have changed the focus of our discussion regarding these results away from the MANET. Accordingly, we have not added comparisons to state-of-the-art routing algorithms.

Figure 4 shows a stretch of 10-50x, which seems quite bad to me, but maybe it isn't.

This figure (in its new version, it is now Figure 4) shows performance of the algorithm for all possible tunings of the following parameter. The high values that the referees state occur for the two extreme cases of complete following (PF=1) or pure random walk (PF=0). Our point is exactly to show that partial following is superior to these and provides a performance that is about 10-fold better. Thus, the fact that the figure depicts the full range of 5-50X strengthens our point of the superiority of random following rather than weakens it.

This particular figure (along with Figure 4) nicely demonstrates how flat the minimum of the stretch function is. In other words, a relatively low stretch of 10 or less can be achieved over a wide range of parameters (between 0.5 and 0.9). In the current context at which this figure is now presented the emphasis is on demonstrating the robustness of the PF model (see the new section titled ‘Robust routing’) rather than on MANETs.